# Temporal FiLM: Capturing Long-Range Sequence Dependencies with Feature-Wise Modulation

**Sawyer Birnbaum**[*12]**, Volodymyr Kuleshov**[*12]**,**
**S. Zayd Enam**[1]**, Pang Wei Koh**[1]**, and Stefano Ermon**[1]
[1]Stanford University, Stanford, CA
[2]Afresh Technologies, San Francisco, CA
{sawyerb,kuleshov,zayd,pangwei,ermon}@cs.stanford.edu

## Abstract

Learning representations that accurately capture long-range dependencies in sequential inputs — including text, audio, and genomic data — is a key problem in deep learning. Feed-forward convolutional models capture only feature interactions within finite receptive fields while recurrent architectures can be slow and difficult to train due to vanishing gradients. Here, we propose Temporal Feature-Wise Linear Modulation (TFiLM) — a novel architectural component inspired by adaptive batch normalization and its extensions — that uses a recurrent neural network to alter the activations of a convolutional model. This approach expands the receptive field of convolutional sequence models with minimal computational overhead. Empirically, we find that TFiLM significantly improves the learning speed and accuracy of feed-forward neural networks on a range of generative and discriminative learning tasks, including text classification and audio super-resolution.

## 1 Introduction

In many application domains of deep learning — including speech [27, 19], genomics [2], and natural language [49] — data takes the form of long, high-dimensional sequences. The prevalence and importance of sequential inputs has motivated a range of deep architectures specifically designed for this data [20, 31, 50].

One of the challenges in processing sequential data is accurately capturing long-range input dependencies — interactions between symbols that are far apart in the sequence. For example, in speech recognition, data occurring at the beginning of a recording may influence the conversion of words enunciated much later.

Sequential inputs in deep learning are often processed using recurrent neural networks (RNNs) [14, 20]. However, training RNNs is often difficult, mainly due to the vanishing gradient problem [3]. Feed-forward convolutional approaches are highly effective at processing both images [33] and sequential data [31, 51, 16] and are easier to train. However, convolutional models only account for feature interactions within finite receptive fields and are not ideally suited to capture long-term dependencies.

In this paper, we introduce Temporal Feature-Wise Linear Modulation (TFiLM), a neural network component that captures long-range input dependencies in sequential inputs by combining elements of convolutional and recurrent approaches. Our component modulates the activations of a convolutional model based on long-range information captured by a recurrent neural network. More specifically, TFiLM parametrizes the rescaling parameters of a

---

batch normalization layer as in earlier work on image stylization [12, 22] and visual question answering [44]. (Table 1 outlines recent work applying feature-wise linear modulation.)

We demonstrate real-world applications of TFiLM in both discriminative and generative tasks involving sequential data. We define and study a family of signal processing problems called *time series super-resolution*, which consists of reconstructing a high-resolution signal from low-resolution measurements. For example, in audio super-resolution, we reconstruct high-quality audio from a low-quality input containing a fraction (15-50%) of the original time-domain samples. Likewise, in genomic super-resolution, we recover high-quality measurements from experimental assays using a limited number of lower-quality measurements.

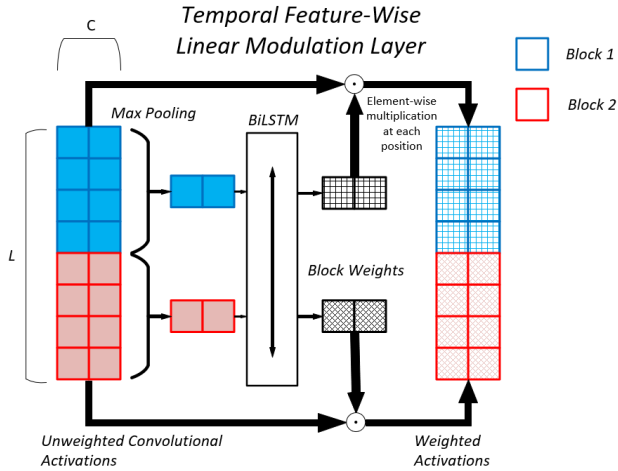

Figure 1: The TFiLM layer combines the strengths of convolutional and recurrent neural networks. *Above*: operation of the TFiLM layer with $T = 8$, $C = 2$, $B = 2$, and a pooling factor of 2.

We observe that TFiLM significantly improves the performance of deep neural networks on a wide range of discriminative classification tasks as well as on complex high-dimensional time series super-resolution problems. Interestingly, our model is domain-agnostic, yet outperforms more specialized approaches that use domain-specific features.

**Contributions.** This work introduces a new architectural component for deep neural networks that combines elements of convolutional and recurrent approaches to better account for long-range dependencies in sequence data. We demonstrate the component's effectiveness on the discriminative task of text classification and on the generative task of time series super-resolution, which we define. Our architecture outperforms strong baselines in multiple domains and could, *inter alia*, improve speech compression, reduce the cost of functional genomics experiments, and improve the accuracy of text classification systems.

## 2  Background

**Batch Normalization and Feature-Wise Linear Modulation.** Batch normalization (batch norm; [23]) is a widely used technique for stabilizing the training of deep neural networks. In this setting, batch norm takes as input a tensor of activations $F \in \mathbb{R}^{T \times C}$ from a 1D convolution layer — where $T$ and $C$ are, respectively, the 1D spatial dimension and the number of channels — and performs two operations: rescaling $F$ and applying an affine transformation. Formally, this produces tensors $\hat{F}, F' \in \mathbb{R}^{T \times C}$ whose $(t, c)$-th elements are the following:

$$\hat{F}_{t,c} = \frac{F_{t,c} - \mu_c}{\sigma_c + \epsilon} \qquad\qquad F'_{t,c} = \gamma_c \hat{F}_{t,c} + \beta_c \qquad (1)$$

Here, $\mu_c, \sigma_c$ are estimates of the mean and standard deviation for the $c$-th channel, and $\gamma, \beta \in \mathbb{R}^C$ are trainable parameters that define an affine transformation.

Feature-Wise Linear Modulation (FiLM) [12] extends this idea by allowing $\gamma, \beta$ to be functions $\gamma, \beta : \mathcal{Z} \to \mathbb{R}^C$ of an auxiliary input $z \in \mathcal{Z}$. For example, in feed-forward image style transfer [12], $z$ is an image defining a new style; by using different $\gamma(z), \beta(z)$ for each $z$, the same feed-forward network (using the same weights) can apply different styles to a target image. [11] provides a summary of applications of FiLM layers.

Table 1: Recent work applying feature-wise linear modulation.

| Paper | Problem Area | Base Modality | Conditioning Modality | Conditioning Architecture |
|---|---|---|---|---|
| Dhingra et al. [9] | QA | Text (document) | Text (query) | CNN |
| Perez et al. [44] and de Vries et al. [7] | Visual QA | Images | Text | RNN/ MLP |
| Dumoulin et al. [12] | Style Transfer | Images | Images | CNN |
| Kim et al. [30] | Speech | Audio | Self (Sequence) | Feedforward |
| Hu et al. [21] | Image Classification | Images | Self | Feedforward |
| This Paper | Sequence Analysis | Audio, Text, DNA | Self (Sequence) | RNN |

---

**Algorithm 1** Temporal Feature-Wise Linear Modulation.

---

**Input:** Tensor of 1D convolutional activations $F \in \mathbb{R}^{T \times C}$ where $T, C$ are, respectively, the temporal dimension and the number of channels, and a block length $B$. **Output:** Adaptively normalized tensor of activations $F' \in \mathbb{R}^{T \times C}$.

1. Reshape $F$ into a block tensor $F^{\text{blk}} \in \mathbb{R}^{B \times T/B \times C}$, defined as $F^{\text{blk}}_{b,t,c} = F_{b \times B + t, c}$.

2. Obtain a representation $F^{\text{pool}} \in \mathbb{R}^{B \times C}$ of the block tensor by pooling together the channels within each block: $F^{\text{pool}}_{b,c} = \text{Pool}(F^{\text{blk}}_{b,:,c})$

3. Compute sequence of normalizers $\gamma_b, \beta_b \in \mathbb{R}^C$ for $b = 1, 2, ..., B$ using an RNN applied to pooled blocks: $(\gamma_b, \beta_b), h_b = \text{RNN}(F^{\text{pool}}_{b,:}; h_{b-1})$ starting with $h_0 = \vec{0}$.

4. Compute normalized block tensor $F^{\text{norm}} \in \mathbb{R}^{B \times T/B \times C}$ as $F^{\text{norm}}_{b,t,c} = \gamma_{b,c} \cdot F^{\text{block}}_{b,t,c} + \beta_{b,c}$.

5. Reshape $F^{\text{norm}}$ into output $F' \in \mathbb{R}^{T \times C}$ as $F'_{\ell,c} = F^{\text{norm}}_{\lfloor t/B \rfloor, t \bmod B, c}$.

---

**Recurrent and Convolutional Sequence Models.** Sequential data is often modeled using RNNs [20, 38] in a sequence-to-sequence (seq2seq) framework [49]. RNNs are effective on language processing tasks over medium-sized sequences; however, on longer time series RNNs may become difficult to train and computationally impractical [51].

An alternative approach to modeling sequences is to use one-dimensional (1D) convolutions. While convolutional networks are faster and easier to train than RNNs, convolutions have a limited receptive field, and a subsequence of the output depends on only a finite subsequence of the input. This paper introduces a new layer that addresses these limitations.

## 3 Temporal Feature-Wise Linear Modulation

In this section, we describe a new neural network component called Temporal Feature-Wise Linear Modulation (TFiLM) that effectively captures long-range input dependencies in sequential inputs by combining elements of convolutional and recurrent approaches. At a high level, TFiLM modulates the activations of a convolutional model using long-range information captured by a recurrent neural network.

Specifically, let $F \in \mathbb{R}^{T \times C}$ be a tensor of activations from a 1D convolutional layer (at one datapoint, i.e. $N = 1$ and we drop the batch size dimension $N$ for simplicity); a TFiLM layer takes as input $F$ and applies the following series of transformations. First, $F$ is split along the time axis into blocks of length $B$ to produce $F^{\text{blk}} \in \mathbb{R}^{B \times T/B \times C}$. Intuitively, blocks correspond to regions along the spatial dimension in which the activations are closely correlated; for example, when processing audio, blocks could be chosen to correspond to audio samples that define the same phoneme. Next, we compute for each block $b$ an affine transformation parameters $\gamma_b, \beta_b \in \mathbb{R}^C$ using an RNN:

$$F^{\text{pool}}_{b,c} = \text{Pool}(F^{\text{blk}}_{b,:,c}) \in \mathbb{R}^{B \times C} \qquad (\gamma_b, \beta_b), h_b = \text{RNN}(F^{\text{pool}}_{b,:}; h_{b-1}) \text{ for } b = 1, 2, ..., B$$

starting with $h_0 = \vec{0}$, where $h_b$ denotes the hidden state, $F^{\text{pool}} \in \mathbb{R}^{B \times C}$ is a tensor obtained by pooling along the the second dimension of $F^{\text{blk}}$, and the notation $F^{\text{blk}}_{b,:,c}$ refers to a slice of $F^{\text{blk}}$ along the second dimension. In all our experiments, we use an LSTM and max pooling.

Finally, activations in each block $b$ are normalized by $\gamma_b, \beta_b$ to produce a tensor $F^{\text{norm}}$ defined as $F^{\text{norm}}_{b,t,c} = \gamma_{b,c} \cdot F^{\text{block}}_{b,t,c} + \beta_{b,c}$. Notice that each $\gamma_b, \beta_b$ is a function of both the current and all the past blocks. Hence, activations can be modulated using long-range signals captured by the RNN. In the audio example, the super resolution of a phoneme could depend on previous phonemes beyond the receptive field of the convolution; the RNN enables us to use this long-range information.

Although TFiLM relies on an RNN, it remains computationally tractable because each RNN is small (the dimensionality of its output is only $O(C)$) and because the RNN is invoked only a small number of times. Consider again the speech example, where blocks are chosen to match phonemes: a 5 second recording contains $\approx 50$ 0.1 second phonemes, yielding only about 50 RNN invocations for 80,000 audio samples at 16KHz. Despite being small, this RNN also carries useful long-range information, as our experiments demonstrate.

## 4 Time Series Super-Resolution

In order to demonstrate the real-world applications of TFiLM, we define and study a new generative modeling task called time series super-resolution, which consists of reconstructing a high-resolution signal $y = (y_1, y_2, ..., y_T)$ from low-resolution measurements $x = (x_1, x_2, ..., x_T)$; $x, y$ denote the source and target time series, respectively. For example, $y$ may be a high-quality speech signal while $x$ is a noisy phone recording.

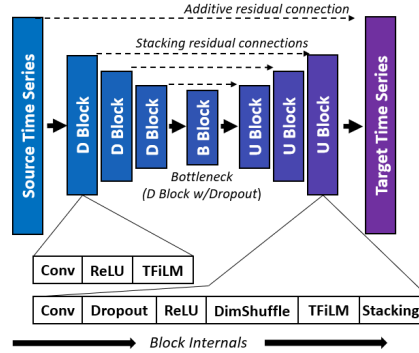

This task is closely inspired by image super resolution [10, 35], which involves reconstructing a high-resolution image from a low-resolution version. As in image super-resolution, we assume that low- and high-resolution time series $x, y$ have a natural alignment, which can arise, for example, from applying a low-pass filter to $y$ to obtain $x$. Below, we provide two examples of time series super-resolution problems.

Figure 2: *Top*: A deep neural network architecture for time series super-resolution that consists of $K$ downsampling blocks followed by a bottleneck layer and $K$ upsampling blocks; features are reused via symmetric residual skip connections. *Bottom*: Internal structure of downsampling and upsampling convolutional blocks.

**Audio Super-Resolution.** Audio super-resolution (also known as bandwidth extension; [13]) involves predicting a high-quality audio signal from a fraction (15-50%) of its time-domain samples. Note that this is equivalent to predicting the signal's high frequencies from its low frequencies. Formally, given a low-resolution signal $x = (x_{1/R_1}, ..., x_{R_1 T/R_1})$ sampled at a rate $R_1/T$ (e.g., low-quality telephone call), our goal is to reconstruct a high-resolution version $y = (y_{1/R_2}, ..., y_{R_2 T/R_2})$ of $x$ that has a sampling rate $R_2 > R_1$. We use $r = R_2/R_1$ to denote the *upsampling ratio* of the two signals. Thus, we expect that $y_{rt/R_2} \approx x_{t/R_1}$ for $t = 1, 2, ..., TR_1$.

**Super-Resolution of Genomics Experiments.** Many genomics experiments can be seen as taking a real-valued measurement at every position of the genome; experimental results can therefore be represented by a time series. Measurements are generally obtained using a large set of probes (e.g., sequencing reads) that each randomly examine a different position in the genome; the genomic time series is an aggregate of the measurements taken by these probes. In this setting, super-resolution corresponds to reconstructing high-quality experimental measurements taken using a large set of probes from noisy measurements taken using a small set of probes. This process can significantly

reduce the cost of scientific experiments. This paper focuses on a particular genomic experiment called chromatin immunoprecipitation sequencing (ChIP-seq) [46].

### 4.1 A Deep Neural Network Architecture for Time Series-Super Resolution

The TFiLM layer is a key part of our deep neural network architecture shown in Figure 2. Other notable features of the architecture include the following: (1) a sequence of downsampling blocks that halve the spatial dimension and double the feature size and of upsampling blocks that do the reverse; (2) max pooling to reduce the size of LSTM inputs; (3) skip connections between corresponding downsamping and upsampling blocks; and (4) subpixel shuffling [47] to increase the time dimension during upscaling and avoid artifacts [41]. For more details, see the Appendix.

We train the model on a dataset $\mathcal{D} = \{x_i, y_i\}_{i=1}^n$ of source/target time series pairs. As in image super-resolution, we take the $x_i, y_i$ to be small patches sampled from the full time series. We train the model using a mean squared error objective.

## 5 Experiments

In this section, we show that TFiLM layers improve the performance of convolutional models on both discriminative and generative tasks. We analyze TFiLM on sentiment analysis, a text classification task, as well as on several generative time series super-resolution problems.

### 5.1 Text Classification

**Datasets.** We use the Yelp-2 and Yelp-5 datasets [1], which are standard datasets for sentiment analysis. In the Yelp-2 dataset, reviews are classified as positive or negative based on the number of stars given by the reviewer. Reviews with 1 or 2 stars are classified as negative, and reviews with 4 or 5 stars are classified as positive. In the Yelp-5 dataset, reviews are labeled with their star rating (i.e., 1-5). The Yelp-2 dataset includes about 600,00 reviews and the Yelp-5 dataset includes 700,00 reviews. For both tasks, there are an equal number of examples with each possible label. With zero-padding and truncation, we set the length of Yelp reviews to 256 tokens.

**Methods.** We tokenize the reviews using Keras's built-in tokenizer and use 100-dimensional GLoVe word embeddings [43] to encode the tokens.

We compare our method against a variety of recent work on the Yelp-2 and Yelp-5 datasets, and to a CNN model based on the architecture of Johnson and Zhang's Deep Pyramid Convolutional Neural Network [25]. This "SmallCNN" model copies the DPCNN architecture, but reduces the number of convolutional layers to 3 and does not include region embeddings. Our full model inserts TFiLM layers after the convolutional layers in this architecture. We normalize the number of parameters between the basic SmallCNN model and the full model that includes TFiLM layers. To do so, we adjust the number of filters in the convolutional layers.

We train for 20 epochs using the ADAM optimizer with a learning rate of $10^{-3}$ and a batch size of 128.

**Evaluation.** Table 2 presents the results of our experiments. On both datasets, TFiLM layers improve the performance of the SmallCNN architecture; the resulting TFiLM model performs at or near the level of state-of-the-art methods from the literature. The models that outperform the TFiLM architecture use several times as many parameters (VDCNN and DenseCNN) or run unsupervised pre-training on external data (DPCNN, BERT, and XLNet). Additionally, the TFiLM model trains several times faster than the SmallCNN model or a comparably-sized LSTM. (See the Appendix for details.) Overall, these results indicate that TFiLM layers can provide performance and efficiency benefits on discriminative sequence classification problems.

### 5.2 Audio Super-Resolution

**Datasets.** We use the VCTK dataset [53] — which contains 44 hours of data from 108 speakers — and a Piano dataset — 10 hours of Beethoven sonatas [40]. We generate low-resolution audio signal from the 16 KHz originals by applying an order 8 Chebyshev type I low-pass filter before

subsampling the signal by the desired scaling ratio. The SINGLESPEAKER task trains the model on the first 223 recordings of VCTK Speaker 1 (about 30 minutes) and tests on the last 8 recordings. In the MULTISPEAKER task, we train on the first 99 VCTK speakers and test on the 8 remaining ones. Lastly, the PIANO task extends audio super resolution to non-vocal data; we use the standard 88%-6%-6% data split.

**Methods.** We compare our method relative to three baselines: a cubic B-spline — which corresponds to the bicubic upsampling baseline used in image super-resolution; a dense neural network (DNN) based on the technique of Li et. al., 2015 [36]; and a version of our CNN architecture without TFiLM layers.

We instantiate our model with $K = 4$ blocks and train it for 50 epochs on patches of length 8192 (in the high-resolution space) using the ADAM optimizer with a learning rate of $3 \times 10^{-4}$. To ensure source/target series are of the same length, the source input is pre-processed with cubic upscaling. We adjust the TFiLM block length $B$ so that $T/B$ (the number of blocks) is always 32. We use a pooling stride and spatial extent of 8. To increase the receptive field of our convolutional layers, we use dilated convolutions with a dilation factor of 2 [56].

Table 2: Text classification on Yelp-2 and Yelp-5 datasets. Methods with * use unsupervised pre-training (unsupervised region embeddings or transformers) on external data and are not directly comparable. Parameter counts exclude models with lower performance. Embedding parameters are not counted.

| Method | Yelp-2 | Yelp-5 | Param |
|---|---|---|---|
| FastText [26] | 95.7% | 63.9% | Linear |
| LSTM [55] | 92.6% | 59.6% | - |
| Self-Attention [37] | 93.5% | 63.4% | - |
| CNN [31] | 93.5% | 61.0% | - |
| CharCNN [58] | 94.6% | 62.0% | - |
| VDCNN [6] | 95.4% | 64.7% | >5M |
| DenseCNN [52] | 96.0% | 64.5% | >4M |
| DPCNN* [25] | 97.36% | 69.4% | >3M |
| BERT* [48][8] | 98.11% | 70.68% | - |
| XLNet* [54] | 98.45% | 72.2% | - |
| SmallCNN (ours) | 78.1% | 61.5% | <1.5M |
| **SmallCNN+TFiLM (full)** | 95.6% | 62.3% | <1.5M |

Including TFiLM layers significantly increases the number of parameters per layer compared with the DNN baseline and the basic CNN architecture. Accordingly, we adjust the number of filters per layer to normalize the parameter counts between the models.

**Metrics.** Given a reference signal $y$ and an approximation $x$, the signal to noise ratio (SNR) is defined as $\text{SNR}(x, y) = 10 \log \frac{||y||_2^2}{||x-y||_2^2}$. The SNR is a standard metric used in the signal processing literature. The log-spectral distance (LSD) [18] measures the reconstruction quality of individual frequencies as follows $\text{LSD}(x, y) = \frac{1}{T} \sum_{t=1}^{T} \sqrt{\frac{1}{K} \sum_{k=1}^{K} \left( X(t, k) - \hat{X}(t, k) \right)^2}$, where $X$ and $\hat{X}$ are the log-spectral power magnitudes of $y$ and $x$, respectively. These are defined as $X = \log |S|^2$, where $S$ is the short-time Fourier transform (STFT) of the signal. We use $t$ and $k$ index frames and frequencies, respectively; we used frames of length 8092.

**Evaluation.** The results of our experiments are summarized in Table 3. According to our SNR metric, our basic convolutional architecture shows an average improvement of 0.3 dB over the DNN and Spline baselines, with the strongest improvements on the SINGLESPEAKER task. Based on the LSD metric, the convolutional architecture also shows an average improvement of 0.5 dB over the DNN baseline and 1.6 dB over the Spline baseline. The convolutional architecture appears to use our modeling capacity more efficiently than a dense neural network; we expect such architectures will soon be more widely used in audio generation tasks.[2]

Including the TFiLM layers improves performance by an additional 1.0 dB on average in terms of SNR and 0.2 dB on average in terms of LSD. The TFiLM layers prove particularly beneficial on the MULTISPEAKER task, perhaps because this is the most complex task and therefore the one which benefits most from additional long-term temporal connections in the model.

Finally, to confirm our results, we ran a study in which human raters assessed the quality of the interpolated audio samples. Our method ranked the best among the upscaling techniques; details are in the appendix.

Table 3: Accuracy evaluation of audio super resolution methods (in dB) on each of the three super-resolution tasks at upscaling ratios $r = 2, 4$, and $8$. DNN and CNN are baselines from the literature. [KEE17] denotes the convolutional method of Kuleshov et al. (2017)

| | | SINGLESPEAKER | | | | MULTISPEAKER | | | | PIANO | | | |
|---|---|---|---|---|---|---|---|---|---|---|---|---|---|
| Ratio | Obj. | Spline | DNN [Li et al.] | Conv [KEE17] | Full Us | Spline | DNN [Li et al.] | Conv. [KEE17] | Full Us | Spline | DNN [Li et al.] | Conv. [KEE17] | Full Us |
| $r = 2$ | SNR | 19.0 | 19.0 | 19.4 | 19.5 | 18.0 | 17.9 | 18.1 | 19.8 | 24.8 | 24.7 | 25.3 | 25.4 |
| | LSD | 3.5 | 3.0 | 2.6 | 2.5 | 2.9 | 2.5 | 1.9 | 1.8 | 1.8 | 2.5 | 2.0 | 2.0 |
| $r = 4$ | SNR | 15.6 | 15.6 | 16.4 | 16.8 | 13.2 | 13.3 | 13.1 | 15.0 | 18.6 | 18.6 | 18.8 | 19.3 |
| | LSD | 5.6 | 4.0 | 3.7 | 3.5 | 5.2 | 3.9 | 3.1 | 2.7 | 3.2 | 3.2 | 2.3 | 2.2 |
| $r = 8$ | SNR | 12.2 | 12.3 | 12.7 | 12.9 | 9.8 | 9.8 | 9.9 | 12.0 | 10.7 | 10.7 | 11.1 | 13.3 |
| | LSD | 7.2 | 4.7 | 4.2 | 4.3 | 6.8 | 4.6 | 4.3 | 2.9 | 4.0 | 3.5 | 2.7 | 2.6 |
| # Params. | | N/A | 6.72e7 | 7.09e7 | 6.82e7 | N/A | 6.72e7 | 7.09e7 | 6.82e7 | N/A | 6.72e7 | 7.09e7 | 6.82e7 |

## 5.3 Chromatin Immunoprecipitation Sequencing

We use histone ChIP-seq data from lymphoblastoid cell lines derived from several individuals of diverse ancestry [29] on the following common histone marks: H3K4me1, H3K4me3, H3K27ac, H3K27me3, and H3K36me3. This dataset contains high-quality ChIP-seq data with a high sequencing depth; to obtain low-quality versions, we artificially subsample 1M reads for each histone mark (out of the full dataset of 100+M reads per mark). This mirrors the setup of Koh et. al., 2016 [32], which introduused a simpler convolutional neural network architecture. The Koh results are the state of the art for this task; we use them as a baseline in this section.

Table 4: Pearson correlation of the model output and the high-quality ChIP-seq signal derived from an experiment with high sequencing depth. [K17] indicates results from Koh et al. (2017); linear method performance is estimated.

| | Pearson Correlation | | | | |
|---|---|---|---|---|---|
| Histone | Input [K17] | Linear [K17] | CNN [K17] | CNN Us | **Full Us** |
| H3K4me1 | 0.37 | 0.41 | 0.59 | 0.79 | 0.81 |
| H3K4me3 | 0.63 | 0.67 | 0.72 | 0.66 | 0.90 |
| H3K27ac | 0.55 | 0.61 | 0.77 | 0.85 | 0.89 |
| H3K27me3 | 0.14 | 0.18 | 0.30 | 0.65 | 0.64 |

Formally, given an input noisy ChIP-seq signal $X \in \mathbb{R}^{k \times T}$, where $k$ is the number of distinct histone marks, and $T$ is the length of the genome, we aim to reconstruct a high-quality ChIP-seq signal $Y \in \mathbb{R}^T$. We use the $k$ low-quality signals as input and train a separate model for each high quality target mark. We use $B = 2$ and training windows of length 1000; all other hyper-parameters are as in the audio-super resolution task.

To evaluate our results, we measure Pearson correlation between our model output and the true, high-quality ChIP-seq signal; this is a standard comparison metric in the field (e.g., [15]). As shown in Table 4, our method significantly improves the quality of the input signal over the Koh results, and on all but one histone mark outperforms the specialized CNN baseline. Across all of the histone marks, the model output from an input of 1M sequencing reads is equivalent in quality to signal derived from 10M to 20M reads, constituting a significant efficiency gain.

## 5.4 Additional Analyses

**Model Visualization.** We examined the internals of the TFiLM layer by visualizing the adaptive normalizer parameters in the audio super-resolution and sentiment analysis experiments. On the former, we observed that the parameters tend to cluster by gender, suggesting that the layer learns useful features. Figures are in the Appendix.

**Ablation Analysis.** The ablation analysis in Figure 5 indicates that temporal adaptive normalization significantly improves model performance. In addition, we performed an ablation analysis for the skip connections and found that they also significantly improve reconstruction accuracy. Our results are in the Appendix.

**Model Generalization.**    We examined the extent to which the model generalizes across datasets. On the audio task, we observed a loss in performance when evaluating the model that was trained on speech on piano music (and vice versa). This highlights the need for diverse training data. Details are in the Appendix.

**Missing Value Imputation.**    We experimented with imputing missing values from a sequence of daily grocery retail sales using various zero-out rates. TFiLM layers consistently provided performance benefits. Our full methodology and results are in the Appendix.

## 6    Previous Work and Discussion

**Feature-Wise Linear Modulation.**    Previous work has applied feature-wise linear modulation to tasks including question answering, style transfer, and speech recognition (see Table 1). Our approach is most similar to that of Kim et al. [31], which modulates layer normalization parameters using a feed-forward model conditioned on an input audio sequence. Conversely, our method adjusts the batch normalization parameters of a feed-forward CNN using an RNN conditioned on the entire sequence. This significantly improves the CNN's performance.

**Time Series Modeling.**    In the machine learning literature, time series signals have most often been modeled with auto-regressive models, of which variants of recurrent networks are a special case [17, 38, 40]. Our approach generalizes conditional modeling ideas used in computer vision for tasks such as image super-resolution [10, 35] or colorization [57].

**Applications to Audio and Genomics.**    Existing learning-based approaches include Gaussian mixture models  [5, 42, 45], linear predictive coding [4], and neural networks [36]. Other recent work on audio super-resolution includes Wang et al.'s WaveNet model [51] and Macartney and Weyde's Wave-U-Net model [39]. Our work proposes a new architecture, which scales better with data size and outperforms recent methods. Moreover, existing techniques involve hand-crafted features [45]; our approach is fully domain-agnostic. Statistical modeling of genomic data has been explored in population and functional genomics [34, 15]; our approach has the potential to make scientific experiments significantly more affordable.

**Computational Performance.**    Our model is computationally efficient and can be run in real time. Unlike sequence-to-sequence architectures, our model does not require the complete input sequence to begin generating an output sequence.

## 7    Conclusion

In summary, our work introduces a temporal adaptive normalization neural network layer that integrates convolutional and recurrent layers to efficiently incorporate long-term information when processing sequential data. We demonstrate the layer's effectiveness on three diverse domains. Our results have applications in areas including text-to-speech generation and sentiment analysis and could reduce the cost of genomics experiments.

**Acknowledgments**

This research was supported by NSF (#1651565, #1522054, #1733686), ONR (N00014-19-1-2145), and AFOSR (FA9550-19-1-0024)

## Footnotes

[2]We also experimented with a sequence-to-sequence architecture. This model preformed very poorly, achieving SNR of about 0 dB for all upscaling ratios. As discussed above, sequence-to-sequence models generally struggle to solve problems involving extremely long time-series signals, as is the case here.

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
