[Supplementary Material]

Table 5: Accuracy evaluation of sentiment analysis methods.

| Experiment | Small (1M Params.) | | | Large | | |
|---|---|---|---|---|---|---|
| Model | SmallCNN | LSTM | **TFiLM** | SmallCNN | LSTM | **TFiLM** |
| Accuracy | 78.1% | 95.2% | 95.6% | 78.0% | 95.2% | 95.3% |
| # Params. | 1.06e6 | 1.03e6 | 1.04e6 | 1.50e6 | 9.61e6 | 2.77e7 |
| Secs. per Epoch | 896 | 1141 | 340 | 1069 | 1655 | 728 |

Figure 3: Learning curves for the 1-million parameter Yelp-2 experiment. *Left*: validation accuracy; *Right*: validation loss. Note that the accuracy and loss converge several epochs slower for the LSTM model compared with the TFiLM model.

## A    Additional Sentiment Analysis Experiments

**Additional Comparisons.**    To measure the memory and run-time efficiency of the TFiLM model, we compare the TFiLM model against the basic SmallCNN architecture and a one-layer LSTM network. We run two experiments, one in which the number of parameters between the models is normalized to about 1 million, and one in which we increase the size of each model so that it uses almost all of the memory of the GPU (a NVIDIA Tesla P100). Note that in the latter experiment the number of parameters varies depending which layer acts as the memory bottleneck.

**Evaluation.**    Table 5 presents the results of our experiments. In keeping with our other findings, in each experiment, the TFiLM model preforms significantly better than the basic SmallCNN architecture. The TFiLM model performs only slightly better than the LSTM model; this is unsurprising, as the sequences are only of length 256, short enough that the pure RNN can avoid the vanishing gradient problem.

Moreover, the TFiLM model trains on average over 50% faster than the SmallCNN model and almost twice as fast as the LSTM model. Figure 3 presents learning curves for the 1-million parameter Yelp-2 experiment. On this experiment, the TFiLM model trains over twice as fast as the SmallCNN model and over three times as fast as the LSTM model.

## B    Time Series Super-Resolution Model Details

**Bottleneck Convolutional Layers**    The core of the model is formed by $K$ successive downsampling and upsampling *layer blocks*: each performs a convolution, dropout, and ReLU non-linearity. Downsampling block $k = 1, 2, ..., K$ contains $\max(2^{6+k}, 512)$ convolutional filters of length $\min(2^{7-k} + 1, 9)$ with a stride of 2. Upsampling block $k$ has $\max(2^{7+(K-k+1)}, 512)$ filters of length $\min(2^{7-(K-k+1)} + 1, 9)$. Thus, at a downsampling step, we halve the spatial dimension and double the filter size; during upsampling, this is reversed. This bottleneck architecture resembles a conovlutional auto-encoder and encourages the model to learn a hierarchy of features.

**Max Pooling.**    Because we expect correlation between data at consecutive time-steps, operating the LSTM over $T/B \times C$ tensors would be inefficient, especially in the first downsampling blocks. We

Table 6: Comparison of audio super-resolution results with a bidirectional LSTM in the TFiLM layer. Switching to a BiLSTM generally provides a minor improvement in performance.

| Experiment | Ratio | Obj. | TFiLM w/LSTM | TFiLM w/BiLSTM | Improvement |
|---|---|---|---|---|---|
| SINGLESPEAKER | 4 | SNR | 16.8 | 16.9 | +0.1 |
|  |  | LSD | 3.5 | 3.6 | -0.1 |
| PIANO | 4 | SNR | 19.3 | 20.5 | +1.2 |
|  |  | LSD | 2.2 | 2.1 | +0.2 |
| MULTISPEAKER | 2 | SNR | 19.8 | 19.6 | -0.2 |
|  |  | LSD | 1.8 | 1.7 | +0.1 |
| MULTISPEAKER | 4 | SNR | 15.0 | 15.1 | +0.1 |
|  |  | LSD | 2.7 | 2.6 | +0.1 |

use max pooling to reduce the size of the LSTM inputs. Specifically, after step 1 of Algorithm 1, we apply max pooling to condense $F^{\text{blk}}_{n,b,t,c}$ tensors into $F^{\text{blk'}}_{n,b,t,c,f,s} = F_{n,((b \times t)-f)/s,c}$ tensors, where $f$ is the pooling spatial extent and $s$ is the pooling stride.

**Skip Connections.** When the source series $x$ is similar to the target $y$, downsampling features will also be useful for upsampling [24]. We thus add additional skip connections that stack the tensor of $k$-th downsampling features with the $(K - k + 1)$-th tensor of upsampling features. We also add an additive residual connection from the input to the final output: the model thus only needs to learn $y - x$. This speeds up training.

**Subpixel Shuffling.** To increase the time dimension during upscaling, we have implemented a one-dimensional version of the subpixel layer of [47], which has been shown to be less prone to produce artifacts [41]. Given a $N \times T \times C$ input tensor, the convolution in a U-block outputs a tensor of shape $N \times T \times C/2$. The subpixel layer reshuffles this tensor into another one of size $N \times 2T \times C/4$; these are concatenated with $C/4$ features from the downsampling stage, for a final output of size $N \times 2T \times C/2$. Thus, we have halved the number of filters and doubled the spatial dimension.

## C  Bidirectional RNN

In some applications – like real-time audio super-resolution – samples from the future may not be accessible; therefore, in our experiments we left the TFiLM RNN uni-directional for full generality. To assess the impact of using a bidirectional RNN, we reran some of our super-resolution experiments with a BiLSTM. As Table 6 shows, in most cases the BiLSTM provides only a minor benefit, and in some cases it even reduces performance (perhaps due to overfitting).

## D  TSNE Embeddings

We generated t-Distributed Stochastic Neighbor Embedding (t-SNE) plots of the adaptive batch normalization parameters on the MULTISPEAKER audio super-resolution task and on the 1-million parameter Yelp review sentiment analysis task. t-SNE is a non-linear dimensionality reduction algorithm that allows one to visualize relationships between the activations on different data points. Figure 4 shows that activations of the final TFiLM layer reflect high-level concepts, including the gender of the speaker and the sentiment of the review.

## E  MUSHRA Test

We confirmed our objective audio super-resolution experiments with a study in which human raters assessed the quality of super-resolution using a MUSHRA (MUltiple Stimuli with Hidden Reference

Figure 4: *Left*: t-SNE plot of activations after the final TFiLM layer for the $r = 4$ model trained on MULTI-SPEAKER recordings. The male speakers (blue) are generally separated from the female speakers (red). *Right*: t-SNE plot of activations after the final TFiLM layer for 1-million parameter Yelp review experiment. The positive reviews (blue) are seperated from the negative reviews (red).

Figure 5: Model ablation analysis on the MULTISPEAKER audio super-resolution task with $r = 4$.

and Anchor) test. For each trial, an audio sample was upscaled using different techniques[3]. We collected four VCTK speaker recordings of audio samples from the MULTISPEAKER testing set. For each recording, we collected the original utterance, a downsampled version at $r = 4$, and signals super-resolved using Splines, DNNs, and our model (six versions in total). We recruited 10 subjects and used an online survey to them to rate each sample reconstruction on a scale of 0 (extremely bad) to 100 (excellent). Table 7 summarizes the results. Our method ranked as the best of the three upscaling techniques.

# F  Additional Ablation Analysis.

Figure 5 displays the result of a longer ablation analysis: the green line displays the validation set $\ell_2$ loss of the original model over time; the yellow curve removes the additive residual connection; the green curve further removes the additive skip connection (while preserving the same total number of filters). This shows that symmetric skip connections are crucial for attaining good performance; additive connections provide an additional small, but perceptible, improvement.

Table 7: MUSHRA test user study scores. We show scores for each sample, averaged over individual users. The average across all samples is also displayed.

| | MULTISPEAKER Sample | | | | |
|---|---|---|---|---|---|
| | 1 | 2 | 3 | 4 | Average |
| Ours | 69 | 75 | 64 | 37 | 61.3 |
| DNN | 51 | 55 | 66 | 53 | 56.3 |
| Spline | 31 | 25 | 38 | 47 | 35.3 |

Table 9: Out-of-distribution performance. We train models on the PIANO and MULTISPEAKER datasets at $r = 4$ and measure SNR and LSD (in dB) on a different testing dataset.

|  | PIANO (TEST) | | MULTISPKR (TEST) | |
|---|---|---|---|---|
|  | SNR | LSD | SNR | LSD |
| PIANO (TRAIN) | 23.5 | 3.6 | 9.6 | 4.1 |
| MULTISPKR (TRAIN) | 0.7 | 8.1 | 16.1 | 3.5 |

# G   Understanding the Generalization of the Super-Resolution Model

We tested the sensitivity of our method to out-of-distribution input via an audio super-resolution experiment in which the training set did not use a low-pass filter, while the test set did, and vice versa. We focused on the PIANO task and $r = 2$. The output from the model was noisier than expected, indicating that generalization is an important concern. We suspect this behavior may be common in super-resolution algorithms but has not been widely documented. A potential solution might be to train on data that has been generated using multiple techniques.

In addition, we examined the ability of our model to generalize from speech to music and vice versa. We found that switching domains produced noisy output, again highlighting the specialization of the model.

Table 9 reports objective metrics for models trained on the MULTISPEAKER and the PIANO tasks and tested both on the same and on the other dataset. Listening to the samples, we found that although the model predicts many high frequencies, these are often corrupted with noise. Thus, our neural networks appear to learn a dictionary that is specialized to the type of audio that they are trained on.

Table 8: Sensitivity of the model to whether low-resolution audio was subject to a low-pass filter (LPF) in dB.

|  | LPF (Test) | | No LPF (Test) | |
|---|---|---|---|---|
|  | SNR | LSD | SNR | LSD |
| LPF (Train) | 30.1 | 3.4 | 0.42 | 4.5 |
| No LPF (Train) | 0.43 | 4.4 | 33.2 | 3.3 |

# H   Missing Data Imputation

We also considered the super-resolution task of imputing missing values in daily retail sales data. Missing values naturally occur in financial time series due to bookkeeping errors or censoring, and they occur in other domains for myriad reasons. Robustness to missing values improves the reliability of downstream machine learning algorithms.

We downloaded publicly available grocery retail sales data from Kaggle's Grocery Sales Forecasting Competition [28]. From this data, we extracted sales figures for $1452$ items on $1024$ days. We split the data 80% / 20% into training and testing sets, and we experiment with setting 10%, 20%, and 30% of the values to zero uniformly at random. We train the model (with and without TFiLM layers) to fill in the missing values. We train for 50 epochs using the ADAM optimizer with a learning rate of $3 \times 10^{-4}$. As in the audio super-resolution tasks, we compare our results with a cubic B-spline and a DNN. (The DNN hyper-parameters are the same as in the audio experiments.)

As Table 10 shows, the convolutional architecture consistently outperforms both baselines, and including TFiLM layers consistently provides an additional benefit.

Table 10: Accuracy evaluation of time series imputation methods (using L2 distance) with zero-out rates of 10%, 20%, and 30%.

| % Missing | Spline | DNN | Conv. | Full |
|---|---|---|---|---|
| 10% | 2.48 | 2.45 | 1.00 | 0.84 |
| 20% | 3.55 | 3.30 | 1.39 | 1.22 |
| 30% | 4.32 | 3.97 | 1.69 | 1.48 |

## Footnotes

[3]We anonymously posted our set of samples to `https://anonymousqwerty.github.io/audio-sr/`. We will release our source code there as well.