[Reviews · NeurIPS 2019]

Reviewer 1



Two or three relevant citations: Transformer models should probably be mentioned in the section on "models designed specifically for use on sequences", since they are competing heavily with the referenced baselines on NLP tasks especially. I believe your numbers on the Yelp dataset compare very favorably to the "sentiment neuron" work from Radford et al https://arxiv.org/abs/1704.01444 - that could be a nice addition and add further external context to your results. However it also worth noting that the ngram results from https://arxiv.org/abs/1509.01626 (which is also a good citation) are quite strong, and also worth noting in the results. Some questions about the architecture, particularly the importance of the "additive skip connection" from input to output - how crucial is this connection, since it somewhat allows the network to bypass the TFiLM layers entirely? Does using a stacked skip (with free trainable parameters) still work, or does it hurt network training / break it completely? What is the SNR of the cubic interpolation used as input for the audio experiments? In other words, what is the baseline SNR of the input, with no modifications? What is the cutoff frequency of the order 8 filter for each super-resolution task? The order is relatively less important than the remaining powerin the "aliasing band" after filtering, which would be related to the cutoff frequency. Specifically for evaluating audio super-resolution, it would also be nice to show numbers for other related work for example http://www1.se.cuhk.edu.hk/~hccl/publications/pub/2018_201811_ISCSLP_MuWANG.pdf , and some other papers such as DRCNN as well as other followups to DRCNN. You cite DRCNN, but don't comapre directly to it from what I can tell. The numbers here appear competitive with what I have seen in the literature, but it would be nice to ground the numbers with some other publications (while also paying attention to whether the datasets match etc. with the publication versions). Given the close relation of the audio model to U-Net, one of the audio U-Net type papers (such as https://openreview.net/forum?id=B1zKGg3soX) would potentially be another strong baseline. As a general note, I am not convinced of the usefulness of SNR as a measure for downsampled audio. Because we know that aliasing implies that *any* sinusoid which matches at the samples points is a possible correct result, the potential mapping from downsampled audio -> upsampled audio is one-to-many, meaning that a result which sounds OK, could have bad SNR compared to the reference. The authors do a MUSHRA test, which is great - maybe this should be included in the main body of the paper, rather than the appendix. AB, ABX, or MUSHRA type tests are better choices in my opinion for testing the quality of the audio upsampling models. This could also be improved by testing against known recent models in the literature, rather than simple baselines of spline and DNN For the genomics experiments, it is again hard to contextualize the correlation numbers, and what they mean for end applications. Are there other models in the literature on this task that you could list in the table, beyond the simple baselines listed? As it stands, I cannot really evaluate the numbers beyond "TFiLM seems better than some CNN, and some LSTM". Having a published result to compare to would again make me more convinced of this result - as it stands this experiment neither helps nor hurts my score of the paper. You mention specifically "can be run in real time" - are there any examples of this? What is the latency you would consider "real time", and what are the primary factors that prevent it from running in real-time today, if any? The last sentence of the conclusion stood out to me - "applications in areas including text-to-speech generation and sentiment analysis and could reduce the cost of genomics experiments" could be better worded as "application to text-to-speech generation, sentiment analysis, and cheaper genomics experiments". Overall, this paper shows strong performance on several benchmarks, and clearly explains its core methodology. The appendix contains a wealth of useful reading, and overall I liked the paper. My chief criticism is in regards to referencing other publications for baselining and performance, as simply testing against your own CNN / DNN / LSTM is not particularly convincing - I had to dig up external references to contextualize the numbers. Having the context directly in the paper would really strengthen the quantitative aspect of this work in my opinion. I hope the authors do release code in the future (as mentioned in the appendix), as this work seems like something the community could build upon in the future. The method described in this work seems straight-forward, and it would be interesting to apply it to a broader range of sequence problems. POST REBUTTAL: The authors addressed my primary concerns with baselining performance clearly with related papers in the results tables, as well as the other critiques of the reviewers, so I improved my score slightly. Overall, I like this paper and will look for implementations (hopefully using author-released code) used in new areas in the future.

Reviewer 2



This paper introduces TFiLM, a temporal extension to Feature-wise Linear Modulation (FiLM) layers which is especially well-suited to sequence modelling tasks. The main contribution is the introduction of a novel architecture that effectively uses convolutional and recurrent layers. Specifically, the authors propose the following processing pipeline: 1) perform a convolutional layer, 2) split the resulting activations in T blocks 3), max-pool the representations within each block, 4) apply an LSTM over the block representations which outputs affine scaling parameters at each step, 5) modulate the representations in each block with the FiLM parameters. The paper evaluates the new architecture on three sequence modelling tasks: text classification, audio super resolution, and chromatin immunoprecipitation sequence super-resolution. For the text classification tasks, the proposed architecture is compared against an LSTM and ConvNet baseline. The proposed architecture always outperforms the pure ConvNet baseline, and improves over the LSTM when the context is long. For the super-resolution tasks, the TFILM architecture consistently outperforms ConvNet baselines. Strengths: - The paper is well-written and easy-to-follow - The proposed architecture is easy to implement and performs well across a wide array of sequence modelling tasks Main concerns: - While the baselines are sensible, the paper doesn’t report the state-of-the-art results for the benchmarks. Reporting SOTA numbers would increase my confidence that you’ve put enough effort into tuning the hyper parameters of the baselines. - The RNN enforces left-to-right processing. I wonder whether two-way processing would further increase performance, for example with a bidirectional RNN or transformer model. Did you experiment with this? Minor: - Missing citations: The self-modulation technique is very related to Squeeze-and-Excitation networks, Hu et al. Also, FiLM layers for VQA were introduced in Modulating Early Visual Processing by Language, de Vries et al. - l. 109 It is only until this line that you specify that you’re using max-pooling. I’d suggest to specify this earlier in the manuscript.

Reviewer 3



The rebuttal addressed my concerns on the baselines for the text classification task. ------ The idea of TFiLM is novel, and the formulation of the module makes a lot of sense. I would give a higher score if the rebuttal can address the following concern on experiments. In the experiments, there is a lack of proper baselines. I am not an expert in audio super-resolution or genome sequencing. That said, at a glance, the audio super resolution only includes self-contained results produced by the authors. It is therefore hard to evaluate the relative significance with respect to that literature. The baselines for text classification seem quite weak, especially for convolutional networks. Also, for text classification it has been shown that beating a bag-of-words model is quite difficult, and the paper did not include benchmarks on them.

[Author Response · NeurIPS 2019]

We thank the reviewers for their constructive feedback. Reviewers found our method to be novel (rev. 1,2,3), clearly
presented (rev. 1,2), sensible and well-motivated (rev. 1,2,3), and having good empirical performance (rev. 1,2,3). The
reviewers' main concern was about the need for additional baselines.

**Additional Baselines.**   To address this main concern,
we are modifying all tables to include results from earlier
publications. On the text classification task, this demon-
strates that we achieve performance competitive with the
SOTA using significantly fewer parameters. On the super-
resolution tasks, we achieve or surpass current SOTA
results.

On the Yelp-2 and Yelp-5 datasets, TFiLM achieves per-
formance competitive with SOTA using fewer parame-
ters (Table 1). The final paper will also include a larger
TFiLM model, attempting to surpass SOTA, and more
datasets (we did not have time to do this for the rebuttal).
We also include the linear FastText baseline.

On the genomics super-resolution task, our method im-
proves over the SOTA results of Koh et al. (2017). This
task was introduced by Koh et al.; Table 2 reports their
baseline, proposed model, our re-implementation of their
model, and our new SOTA result.

On the audio-super resolution tasks, our two existing base-
lines already correspond to the DNN-based method of Li et
al. (2015) (we re-implemented it) and the CNN-based method
from Kuleshov et al. (2017) (using the provided source code).
Our new Table 3 reflects this comparison to standard models.
Note also that we already report the results of the cubic spline
(interpolation) baseline.

Additional baselines are difficult to add, since there is no stan-
dard audio super-resolution benchmark. DRCNN is an image
super-resolution method and its extension to audio is outside of
the scope of our paper. The Wavenet paper cited by Reviewer
1 only performs two single-speaker experiments and uses a different experimental setup, that we didn't have time to
reproduce. We anticipate our performance to be competitive but somewhat lower (they report an LSD of 2.5; our
single-speaker experiment has 3.4; their CNN baselines are 4.0 and 4.5). The U-Net baseline cited by Reviewer 1 is
relevant, but almost identical to our "CNN" baseline (Kuleshov et al., 2017). We will cite all of these papers and we
thank Reviewer 1 for bringing them to our attention.

**Left-to-Right Processing.**   Reviewer 2 is right that TFiLM
can use a bidirectional RNN. In some applications – like real-
time audio super-resolution – samples from the future may not
be accessible; therefore, we left the RNN uni-directional for
full generality. However, we agree that using a bidirectional
RNN is better for presentation, and will do so in the paper.

**Architecture Questions.**   The effects of removing the addi-
tive skip connection are shown in Figure 5. The model trains
much more slowly and achieves somewhat lower performance.
Bypassing the TFiLM layer would revert to a pure Spline model,
whose performance we report. We also report the performance of cubic interpolation, which is the same as "[cubic]
Spline". In our experiments, we were able to run audio super-resolution inference faster than real time, using <1sec for
>30sec of audio in <1sec; we will add more detailed analysis in the final paper.

**Missing Citations.**   We have added a Transformer baseline and a citation. We have also added citations to audio
super-resolution papers (including the Wavenet one). We thank Reviewer 2 for brining the Squeeze-and-Excitation
paper to our attention; we are citing it, as well as FiLM for VQA.

| Method | Yelp-2 | Yelp-5 | Param |
|---|---|---|---|
| FastText [Grave et al., 2017] | 95.7% | 63.9% | Linear |
| LSTM [Yogatama et al., 2017] | 92.6% | 59.6% | - |
| Self-Attention [Lin et al., 2017] | 93.5% | 63.4% | - |
| CNN [Kim, 2014] | 93.5% | 61.0% | - |
| CharCNN [Zhang et al., 2015] | 94.6% | 62.0% | - |
| VDCNN [Conneau et al., 2017] | 95.4% | 64.7% | >5M |
| DenseCNN [Wang et al., 2018] | 96.0% | 64.5% | >4M |
| DPCNN* [Rie, Johnson, 2017] | 97.36% | 69.4% | >3M |
| BERT* [Devlin et al., 2018] | 98.11% | 70.68% | - |
| **SmallCNN (ours)** | 78.1% | 61.5 | <1.5M |
| **SmallCNN+TFiLM (ours)** | 95.6% | 62.3 | 1.5M |

Table 1: Text classification on Yelp-2 and Yelp-5 datasets. Methods with * use unsupervised pre-training (unsupervised region embeddings or transformers) on external data and are not directly comparable. Parameter counts exclude models with lower performance. Embeddings are not counted.

| Histone | Input [K17] | Linear [K17] | CNN [K17] | CNN Us | **Full Us** |
|---|---|---|---|---|---|
| H3K4me1 | 0.37 | 0.41 | 0.59 | 0.79 | 0.81 |
| H3K4me3 | 0.63 | 0.67 | 0.72 | 0.66 | 0.90 |
| H3K27ac | 0.55 | 0.61 | 0.77 | 0.85 | 0.89 |
| H3K27me3 | 0.14 | 0.18 | 0.30 | 0.65 | 0.64 |

Table 2: Genomic super-resolution. [K17] indicates results from Koh et al. (2017); linear method performance is estimated.

| Ratio | Obj. | Spline | DNN [Li et al.] | Conv [KEE17] | **Full Us** |
|---|---|---|---|---|---|
| $r = 2$ | SNR | 18.0 | 17.9 | 18.1 | 19.8 |
|  | LSD | 2.9 | 2.5 | 1.9 | 1.8 |
| $r = 4$ | SNR | 13.2 | 13.3 | 13.1 | 15.0 |
|  | LSD | 5.2 | 3.9 | 3.1 | 2.7 |
| $r = 8$ | SNR | 9.8 | 9.8 | 9.9 | 12.0 |
|  | LSD | 6.8 | 4.6 | 4.3 | 2.9 |

Table 3: Audio super-resolution. DNN and CNN are baselines from the literature. [KEE17] denotes the convolutional method of Kuleshov et al. (2017)

[Meta-Review · NeurIPS 2019]

Post rebuttal all reviewers converged to a solid accept rating.